# Advances in Biomarker-Driven Targeted Therapies in Thyroid Cancer

**DOI:** 10.3390/cancers13246194

**Published:** 2021-12-09

**Authors:** Prachi Mishra, Dipranjan Laha, Robert Grant, Naris Nilubol

**Affiliations:** Surgical Oncology Program, Center for Cancer Research, National Cancer Institute, National Institutes of Health, Bethesda, MD 20892, USA; prachi.mishra@nih.gov (P.M.); dipranjan.laha@nih.gov (D.L.); rob.grant@nih.gov (R.G.)

**Keywords:** thyroid cancer, biomarker, targeted therapy, tyrosine-kinase inhibitors

## Abstract

**Simple Summary:**

This article reviews current treatment practices for thyroid cancer with a focus on novel targeted molecular therapy. Rapidly expanding knowledge of the molecular biology of these cancers coupled with the increased availability of genetic testing has led to exciting paradigm shifts in treatment strategies for these tumor types. We aim to provide up-to-date information on these state-of-the-art therapies as a guide for clinicians who specialize in the treatments of thyroid cancer.

**Abstract:**

Thyroid cancer is the most common type of endocrine malignancy comprising 2–3% of all cancers, with a constant rise in the incidence rate. The standard first-line treatments for thyroid cancer include surgery and radioactive iodine ablation, and a majority of patients show a good response to these therapies. Despite a better response and outcome, approximately twenty percent of patients develop disease recurrence and distant metastasis. With improved knowledge of molecular dysregulation and biological characteristics of thyroid cancer, the development of new treatment strategies comprising novel targets has accelerated. Biomarker-driven targeted therapies have now emerged as a trend for personalized treatments in patients with advanced cancers, and several multiple receptor kinase inhibitors have entered clinical trials (phase I/II/III) to evaluate their safety and efficacy. Most extensively investigated and clinically approved targeted therapies in thyroid cancer include the tyrosine receptor kinase inhibitors that target antiangiogenic markers, BRAF mutation, PI3K/AKT, and MAPK pathway components. In this review, we focus on the current advances in targeted mono- and combination therapies for various types of thyroid cancer.

## 1. Introduction

The worldwide incidence of thyroid cancer in recent decades has steadily risen by almost 20% from 1990 to 2013. Thus, thyroid cancer remains the most common endocrine malignancy [1]. An analysis of the Surveillance Epidemiology and End Results (SEER) database showed a yearly average increase of 3.6% from 1974–2013 in the United States [2]. Recent data have shown that the rise in the incidence of thyroid cancer occurs in all stages, and tumor size and is not solely due to the increased detection of small asymptomatic lesions from more frequent use of cross-sectional imaging, as was hypothesized previously [3].

There are different sub-classifications of thyroid cancer based on histopathology and molecular profiling. Well-differentiated thyroid carcinomas (WDTC) are the most common and have the best outcomes if treated early. Within this group are papillary thyroid carcinoma (PTC), which comprises about 85% of all thyroid cancers, follicular thyroid carcinoma (FTC), and Hurthle cell thyroid carcinoma. Anaplastic thyroid carcinoma (ATC) is the most lethal, very rare, and is considered an undifferentiated subtype. Poorly differentiated thyroid carcinoma (PDTC) represents about 5–10% of all thyroid cancers of follicular cell origin and contains several aggressive histologic subtypes. PDTC is a distinct entity that falls between undifferentiated ATC and WDTC both histologically and clinically. Medullary thyroid carcinoma (MTC), representing about 5% of all thyroid cancers, originates from the parafollicular cells (or C cells), which are neural crest cells in origin [4]. MTC is a distinct entity from cancers of follicular cell origin because it arises from a different cell type, and thus does not fall within any of the above classifications based on the degree of differentiation. While not universal, most WDTCs respond well to surgery with or without radioactive iodine (RAI) therapy with an excellent prognosis, and PDTCs and ATCs are uniformly resistant to RAI treatment and have high rates of locoregional recurrence and distant metastasis after surgery, which contribute to the associated high mortality, especially in ATC. Despite having the best prognosis of thyroid cancers of follicular cell origin, PTC can still metastasize to the central cervical lymph nodes in approximately 50% of cases [5], while PDTC and ATC frequently exhibit hematogenous systemic spread. At a later stage of the advanced disease, systemic treatment, such as small molecule inhibitors, following surgical removal and radioiodine therapy seems to be inevitable. A better strategy to improve therapeutic efficacy in iodine-refractory, aggressive thyroid cancer is essentially needed; therefore, there are currently several active investigations on new targetable molecules or signaling pathways in metastatic MTCs and radioiodine-refractory thyroid cancer of follicular cell origin. Several small molecule inhibitors have been approved by the FDA for different subtypes of thyroid cancer, and a few more are in the pipeline. In this review, we aimed to provide the insight into the molecular alterations with a focus on the targetable biomarkers for mono- and combination systemic therapies for various types of thyroid cancer to date. The summary of current and investigational systemic treatments in thyroid cancers is provided.

## 2. Genetic and Phenotypic Background of Thyroid Cancers

Based on treatment guidelines from the American Thyroid Association (ATA) from 2015, which is the most recent version, there is no specific recommendation to perform next-generation sequencing on any patient given the limitations of these testing panels and the limited clinical utility given that most thyroid cancers are cured with surgery after diagnosis [6]. However, thyroid cancer has different subtypes based on its distinct genetic and epigenetic alterations, driven by a set of oncogenic drivers. Extensive cancer genome and exome profiling of large thyroid cancer databases such as The Cancer Genome Atlas (TCGA) and the Memorial Sloan–Kettering Cancer Center cohort revealed a wide spectrum of somatic mutations, genetic fusions, and copy number alterations that clearly distinguishes WDTC from PDTC [7]. Knowledge of molecular genetics has greatly expanded over the past six years since the ATA guidelines were published and targeted therapies of specific gene and protein alterations have been developed.

### 2.1. Well-Differentiated Papillary and Follicular Thyroid Carcinoma

The well-differentiated PTCs, which account for about 85% of thyroid cancers, are comparatively clinically indolent with a lower number of alterations in their genome. Whole-genome sequencing studies show low mutation densities in these tumors. Nevertheless, thyroid cancer of follicular cell origin encompasses several tumor types by the mutually exclusive driver mutations such as *BRAF* V600E-mutated tumors (60%), non-V600E *BRAF* mutations including K601E, *RAS*-mutated tumors (15%), and mutations or fusions of other receptor tyrosine kinases (12%) such as *RET*, *NTRK* and *ALK* [8]. These driver genes are associated with distinct histologic phenotypes. The histology of thyroid cancer with *BRAF*V600E mutation includes classic PTC and tall cell variant, which is more aggressive than classic PTC and exhibits frequent extrathyroidal extension and lymph node metastasis. PTC with *RAS* mutations are common follicular variants of PTC, a distinct and separate entity from FTC, and they less frequently progress to extrathyroidal extension or lymph node metastasis than either classic or tall-cell-variant PTC [9]. The somatic *RAS* mutations can be observed in FTC with more frequent hematogenous metastasis. PTC with a *BRAF*V600E mutation and PTC with *RET/PTC* rearrangements were both often associated with stage III/IV disease and recurrence, with the *RET* rearrangement oncogene additionally associated with higher distant metastasis [10].

In the pediatric population, WDTC is a common malignancy, particularly among females in their mid to late teen years. Tumors in these patients are found to more commonly contain somatic *RET* fusion events than WDTC lesions in adults. As described above, upwards of half of the adult PTCs harbor *BRAF* mutations—which was thought to be a rare event in the pediatric cohort. However, recent analysis has shown they may contain more *BRAF* mutations than previously reported [11]. Patients with iodine-refractory WDTC and *RET* fusion genes may benefit from RET-targeted therapy.

### 2.2. Poorly Differentiated Thyroid Carcinoma and Anaplastic Thyroid Carcinoma

Of the thyroid cancers of follicular cell origin, PDTC is more aggressive than PTC or FTC, and ATC is the most aggressive with the worst prognosis. Because the ATC cells are poorly differentiated or undifferentiated, they commonly have low or no expression of the membranous glycoprotein: sodium–iodine symporter. A decreased or total lack of expression of this protein results in poor iodine uptake which renders radioactive iodine treatment ineffective and can lead to decreased synthesis of thyroglobulin, the precursor protein to the active thyroid hormones. These de-differentiated cancers tend to be less responsive to thyroid-stimulating hormone (TSH) suppression treatment due to low expression of TSH receptors. PTDCs and ATCs are often seen with the preexisting or previously removed well-differentiated thyroid cancer and therefore share the same driver mutations [12]. In advanced thyroid cancer, *BRAF* mutations were less frequent, whereas the RAS mutations were more so in comparison to the PTCs. Interestingly, genomics and transcriptomics data show that, compared to PTDCs, ATCs had a greater mutation burden, with higher frequencies of *TP53*, *TERT* promoter, PI3K/AKT/mTOR pathway effectors, SWI/SNF and histone methylases. While *EIF1AX* mutation is markedly high in PTDC, ATC dominates in tumors with RAS mutation [7]. Some key genomic alterations distinguish PTDCs from ATCs, which include certain genetic defects such as switch/sucrose non-fermenting (SWI/SNF) complexes and histone methyltransferases (HMTs). They also had distinct clinical phenotypes, such as *BRAF* mutants primarily developing locoregional nodal metastasis, whereas *RAS* mutants were commonly observed with distant metastases. Moreover, ATCs are extensively infiltrated by tumor-associated macrophages (TAMs), and gene sets that define M2 macrophages can clearly separate ATCs from PTDCs [7].

Pediatric PDTC and its oncogenic origins are not well understood. In the mid-2000s, diagnostic criteria for this disease were established with what is called the Turin criteria and are based on histopathologic features. These tumors are exceedingly rare in children and work on their molecular genetics is very scarce. Recent work has shown that in children these tumors can exhibit a somatic *DICER1* mutation, a gene that encodes an RNase involved in processing microRNAs and creating siRNA [13].

### 2.3. Medullary Thyroid Carcinoma

Approximately 75% of MTC occurs sporadically and 25% is hereditary cancer associated with multiple endocrine neoplasia type 2 (MEN2) syndromes or familial medullary thyroid cancer (FMTC) syndrome [14,15]. In almost all cases of these syndromes, *RET* (re-arranged during transfection) proto-oncogene germline mutations are implicated [15]. MEN2A is characterized by MTC, pheochromocytoma, and primary hyperparathyroidism (pHPT), and is sometimes associated with either cutaneous lichen amyloidosis in patients with *RET* codon 634 mutation or Hirschsprung’s disease. The severity of MTC, which eventually develops in virtually all patients with MEN2A, and the presence of other phenotypic features depend on which codon within the *RET* gene is the culprit. Which codon is mutated does not seem to change the penetrance of MTC in MEN2A patients but does change the risk profile and can predict a more aggressive form of MTC with a worse prognosis. Patients with *RET* M918T mutation harbor the highest risk, followed by A883F and mutations to codon 634, which are considered high-risk [15]. All other *RET* mutations are considered moderate risk for MTC. MEN2B is characterized by MTC, pheochromocytoma, and mucosal ganglioneuromas. MTC in MEN2B is more aggressive than in MEN2A as MTC frequently presents in early childhood with lymph node metastasis and is caused by the germline *RET* codon M918T mutation in 95% of patients [15]. The current consensus as outlined in the 2015 Revised American Thyroid Association Guidelines for the Management of MTC is that FMTC is not a distinct clinical entity, but rather a variant of MEN2A exhibiting a phenotype that presents with only MTC, without pHPT or pheochromocytoma [15]. Activating somatic *RET* mutations have been implicated in the pathogenesis of sporadic MTC and are seen in approximately 50% of cases [15,16]. In non-neoplastic normal healthy tissue, ligand binding triggers dimerization followed by activation of the RET receptor which, induces the cell proliferation machinery. In cancer cells, the point mutations in the codons in the *RET* oncogene outlined above cause autophosphorylation of tyrosine residues leading to a constitutive activation of the RET receptor and permanent gain of function [16,17].

## 3. Treatment Strategies for Thyroid Cancer

### 3.1. Surgical Intervention

Indications to perform surgery for a thyroid lesion depend on radiographic and cytologic features which help determine the risk of malignancy of a thyroid nodule, as well as family history and symptomatology. A thyroid nodule that meets size criteria and has certain features on ultrasound will undergo fine needle aspiration (FNA) and cytologic evaluation. Based on FNA results, the lesion will be stratified by the Bethesda System for Reporting Cytopathology, which places the nodule into one of six categories of increasing likelihood of malignancy [6]. The standard of care for a diagnosis of high suspicion of malignancy of a lesion greater than 1 cm or malignancy of any size on FNA, Bethesda V, and VI, respectively, is thyroidectomy (TT). However, the extent of thyroidectomy continues to be controversial in patients with low-risk differentiated thyroid cancer. The established standard of care that provides excellent outcomes is total thyroidectomy [6,18]. Because most patients with low-risk differentiated thyroid cancer have excellent prognosis regardless of the extent of thyroidectomy, there is an increasing body of literature supporting thyroid sparing surgery in the form of ipsilateral thyroid lobectomy (TL) for small node-negative PTC smaller than 4 cm, which shows equivalent long-term outcomes to TT [19,20,21]. For Bethesda III and IV lesions, which identify atypical cells on FNA, but which are not diagnostic of malignancy, TL can often be performed to spare the patient life-long thyroid hormone replacement [6,18]. For asymptomatic lesions that fall into Bethesda II, the benign category, non-operative management is recommended [6]. Bethesda I is non-diagnostic, and in most cases repeat FNA is indicated [6]. These recommendations are not specific to which cancer is being treated. For all cancers of follicular cell origin, central neck lymph node dissection should also be performed for clinically positive nodes identified during surgery or on ultrasound, and lateral neck dissection should be performed for biopsy-proven positive lateral lymph nodes [6]. Because radioiodine treatment is an effective systemic targeted therapy in patients with metastatic differentiated thyroid cancer, the primary tumors and the remainder of the thyroid gland should be removed to facilitate the use of radioiodine treatment.

In syndromic patients at risk for MTC, the recommendation is prophylactic TT at a young age. Patients with MEN2A with high-risk *RET* mutations should have their thyroid removed by age 5; patients with moderate risk MEN2A should have TT when serum calcitonin is elevated; in MEN2B, TT is indicated in the first year of life [15].

The decision-making process for surgical management of ATC is significantly more complex than other cancers of follicular cell origin or MTC primarily because of the poor survival and the involvement of surrounding neck structures. While patients who have surgery live significantly longer than patients who do not (6.6–8 vs. 2.1–3 months median survival), outcomes are still abysmal [22,23]. Several factors must be taken into account when deciding to proceed with surgery, including the need for urgent or emergent palliative surgery to relieve airway compromise, other treatment options including neo-adjuvant or adjuvant systemic therapy, the patient’s goals of care, and the ability to achieve a microscopically negative margin (R0 resection).

### 3.2. Radioactive Iodine Therapy

Radioactive iodine ablation therapy is indicated in certain cases of WDTC because most differentiated thyroid cancers of follicular cell origin preferentially and avidly uptake iodine. RAI therapy is used in the adjuvant setting in patients with incomplete resection, locally advanced disease, or who develop distant metastasis. The ATA classifies WDTC as low, intermediate, or high risk, which is based on various tumor features, including stage, micro or macroscopic invasion, vascular invasion, and completion of resection, among others. There is little evidence that in early-stage and low-risk WDTC RAI adds any disease-specific survival benefit after surgery [6,24]. Patients with intermediate-risk WDTC may have improved disease-free survival, but RAI is not recommended pro forma in all patients [6]. There are observational data showing both overall and disease-free survival benefits from adjuvant RAI in high-risk thyroid cancers, which is indicated routinely in these cases [6].

RAI treatment was first studied in the late 1930s at the Massachusetts Institute of Technology and The Massachusetts General Hospital. They initially showed that iodine isotopes were readily taken up by the thyroid gland and quickly adapted this application to the treatment of hyperthyroidism followed by thyroid cancer in the 1940s [25]. In the modern use of RAI, the patient is given iodine-131 radioisotope, which is taken up by thyroid follicular cells via the sodium–iodine symporter in the same process of uptake as natural iodine, which is the precursor to thyroglobulin and thyroid hormone [26]. Thyroid glandular tissue and differentiated thyroid cancer cells preferentially take up nearly all the iodine in circulation, which is targeted. The primary goal of this therapy is to cause radiation-induced cytotoxicity in all residual thyroid tissue and thyroid cancer to simultaneously reduce tumor recurrence and to facilitate surveillance. By ablating all residual thyroid tissue, recurrence can be monitored by checking thyroglobulin levels, which would be undetectable after successful ablation and increase the setting of recurrence. Additionally, the therapy is also more effective for the *RAS*-mutated PTCs than *BRAF*-mutated tumors, possibly because the *BRAF* mutation reduces the expression of genes that are required for iodine transport and metabolism ultimately reducing the radioiodine uptake [27].

### 3.3. Chemotherapy

Cytotoxic chemotherapy is rarely indicated for WDTC. Although the efficacy is limited, the chemotherapeutic drugs often used in *BRAF* wild-type ATCs are paclitaxel and docetaxel, the anthracyclines (doxorubicin), and the platins such as cisplatin and carboplatin, frequently in combination with external radiotherapy [22]. A combination of taxanes and platins was observed to be more effective than the single drug alone [28,29]. Some of the cytotoxic agents, such as gemcitabine and vinorelbine, demonstrated better response in ATC cell lines but never entered clinical trials due to their inefficacy in controlling the ATC progression [30]. In an open-label clinical trial of carboplatin/paclitaxel chemotherapy with or without fosbretabulin (a microtubule destabilizing drug, targeting vasculature) in ATC patients, the median OS for CP/fosbretabulin was 5.2 months compared to CP alone, which was 4 months with a one-year survival of 26% vs. 9% in patients who received CP/fosbretabulin and CP only, respectively [31]. The study was withdrawn as the study did not meet the significant improvement of OS criteria due to low accrual. The lack of durable anti-tumor efficacy of the current cytotoxic chemotherapy for *BRAF-*WT ATC emphasizes an urgent need to identify novel effective treatments. More than 50 clinical trials in thyroid cancer are currently underway with a combination approach of chemotherapy drugs with targeted or immunotherapy drugs.

## 4. Targeted Therapies in Thyroid Cancer

The improved insight into clinically relevant dysregulated pathways in thyroid cancer is critically important to identify effective treatment regimens, to personalize treatments based on molecular profile, and to risk-stratify patients for an appropriate disease surveillance. With the discoveries of molecular mechanisms associated with thyroid cancer initiation and progression, a new generation of drugs has been designed to target these dysregulated pathways preferentially expressed in thyroid cancer. Several targeted anticancer agents have emerged as promising monotherapy options which include some of the recently evaluated small molecule inhibitors such as targeted inhibitors of tyrosine kinase, immune checkpoint (PDL-1), NOTCH1, and mTOR pathways [32,33,34,35]. New targeted therapies that can block the functions of molecules involved in the oncogenic transformation of indolent tumors to the aggressive ones are among the promising treatment modalities. These novel targets demonstrated significant therapeutic efficacy in preclinical and some clinical platforms and are interesting to follow. We comprehensively described the major groups of targeted therapy in thyroid cancer below.

### 4.1. Tyrosine Kinase Inhibitors (TKIs)

One of the most important kinds of small molecule inhibitor and the most studied inhibitor in thyroid cancer is tyrosine kinase inhibitors (TKIs). The receptor tyrosine kinases (RTKs) play an important role in multiple regulatory processes in normal and cancer cells because of their involvement in transmembrane signaling. RTKs are activated after binding to their ligands, undergo dimerization and autophosphorylation of their cytoplasmic domains, and activate the downstream signaling cascade [36]. RTKs play an important role in cancer initiation and progression. Several TKIs have been developed and evaluated for cancer treatment including thyroid cancers (Table 1). TKIs competitively block the ATP-binding site of the tyrosine kinase domain and hamper their function, thereby blocking the downstream signaling pathways. The majority of differentiated or undifferentiated *BRAF*V600E and P53 mutated thyroid cancer led to an activation of the MAPK pathway, one of the prime pathways involved in the progression of the disease. The progression of such tyrosine kinase-led oncogenic pathways resulting in tumor progression can be suppressed by using multiple-targeted TKIs. Some of the TKIs we described here have been extensively investigated in thyroid cancer preclinical models as well as in clinical trials.

#### 4.1.1. Anti-Angiogenic TKIs

The anti-angiogenic TKIs mainly target the VEGF pathway. A well-known orally active TKI is sorafenib, which has multiple targets such as BRAF, VEGFR1, VEGFR2, and RET and causes apoptosis and exhibits anti-angiogenic effects in vivo. In the preclinical model of mice with human ATC xenografts, sorafenib inhibited tumor growth and improved animal survival. The clinical efficacy of sorafenib was tested in numerous clinical trials, especially for ATC and the metastatic disease, either through monotherapy or in combination with mTOR inhibitors, such as everolimus and temsirolimus. A multi-institutional phase II clinical trial of sorafenib in patients with ATC exhibited modest efficacy with a partial response in 2 out of 20 patients (10%), with five out of 20 patients (25%) exhibiting stable disease. The overall median progression-free survival (PFS) time and median survival were 1.9 and 3.9 months, respectively, with manageable expected toxicity, such as hypertension and skin rash [37]. The multicenter, randomized, double-blind, placebo-controlled phase III trial (DECISION) in patients with advanced/metastatic, iodine-refractory, progressive, differentiated thyroid cancer showed significant improvement in the median PFS (10.5 months) in sorafenib (400 mg orally twice daily) treatment group compared to placebo-treated group (5.8 months) [38], suggesting the drug is effective in patients with progressive, radioactive iodine-refractory DTC.

Sunitinib is an orally available, multi-targeted TKI targeting mainly PDGFRs and VEGFRs, that are involved in angiogenesis and tumor cell proliferation [39]. Sunitinib causes a reduction in tumor vascularization and cancer cell death, leading to tumor shrinkage in patients [39]. A phase II clinical trial to access the efficacy of sunitinib in patients with 18F-FDG-PET avid, iodine refractory WDTC, and MDC was carried out by the administration of 37.5 mg daily on a continuous basis. Among the 35 enrolled patients (7 MTC; 28 WDTC), 33 were evaluable for the disease response. Clinical benefit, such as a complete tumor response, a partial response and a stable disease, was seen in 3%, 28% and 46%, respectively. The median time to progression in patients with progressive disease was 12.8 months, with anticipated TKI-related common toxicities, including palmar-plantar erythrodysesthesia, fatigue, diarrhea, leukopenia, and neutropenia [40]. In a similar multicenter phase II study, 71 patients (41 advanced radioactive iodine resistant DTC, 4 advanced ATC, and 26 MTC) were enrolled between August 2007 to October 2009. The primary endpoint of objective response rate was met in advanced DTC (22%) and MTC (38.5%) patients. Unfortunately, patients with ATC experienced no objective response. Frequent side effects observed were asthenia/fatigue, mucosal, cutaneous toxicities, hand–foot syndrome, including nine unexpected side effects and five induced deaths in patients. Further studies on this inhibitor are hence warranted.

Another multi-targeted anti-angiogenic TKI is lenvatinib, which is an oral drug acting on the angiogenic as well as fibroblast markers, i.e., VEGFR1, VEGFR2, VEGFR3, PDGFRβ, RET, KIT, fibroblast growth factor receptor 1 (FGFR1), FGFR2, FGFR3, and FGFR4. The action of lenvatinib is different than most other TKIs as it mainly abrogates cell migration and invasion rather than affecting the proliferation of tumor cells. There are several ongoing clinical trials currently in phase II, many of which are tested in multiple combinations with other drugs, including Iodine-131. Lenvatinib (E7080) resulted in a longer OS in patients with metastatic, iodine-refractory, differentiated thyroid cancer to the lungs of >1.0 cm, in a random multicenter, randomized double-blind placebo-controlled trial (SELECT) of in 131I-refractory in DTC, which was completed in 2015. The study suggested early initiation of lenvatinib may improve the outcome in radioiodine refractory DTCs [41]. It is approved by both EMA and FDA at a dose of 24 mg/day for use in patients with metastatic and radioiodine-refractory differentiated thyroid carcinoma [42]. Further combinatorial studies of sunitinib and pembrolizumab (checkpoint inhibitor) in locally advanced and metastatic ATCs are currently going on [43]. Imatinib is an oral inhibitor targeting the ABL kinase, KIT receptor, and platelet-derived growth factor receptor (PDGFR). Imatinib has been approved by the Food and Drug Administration (FDA) for the treatment of chronic myelogenous leukemia (CML) and gastrointestinal stromal tumors (GIST) expressing KIT. Preclinical studies have demonstrated dramatic cytostatic effects imatinib on the ATC cells that overexpress ABL kinase in which *TP53* is mutated or deficient. Imatinib showed selective inhibition of ABL kinase activity in these cell lines [44,45]. Imatinib was well tolerated in a phase II clinical trial of patients with ATC and post-treatment of chemoradiation or surgery and developed recurrent or metastatic disease. The trial showed a 6-month progression-free survival (PFS) in 36% of patients and OS in 45% of patients [46]. Evaluation of imatinib in larger clinical trials for ATC is still awaited.

Pazopanib is an orally available, multitargeted inhibitor with in vitro inhibition of VEGFR, PDGFR c-kit and multiple cytochrome P450 enzymes resulting in anti-angiogenesis and the inhibition of several oncogenic signaling pathways [47,48]. The drug is currently approved for advanced renal cell carcinoma and soft tissue sarcomas of patients who have received prior chemotherapy. Randomized phase I and phase II trials are currently ongoing in patients with advanced thyroid cancer to study the adverse effects and efficacy of pazopanib either as monotherapy or in combination with iodine-131. In a previous phase II trial that was carried out in 2008 of pazopanib (800 mg daily in 4-week cycles) in DTC patients, partial responses were observed in 18 out of 37 enrolled patients (response rate of 49%). Two-thirds of patients experienced a durable response greater than one year [49].

Another potent selective inhibitor of VEGF that facilitates the inhibition of angiogenesis and the restriction of tumor blood flow is axitinib. In a phase II trial, axitinib was well tolerated in patients with various tumor histologic subtypes of advanced thyroid cancer that were refractory or not amenable to iodine-131, demonstrating durable responses and long overall survival. Patients who received axitinib experienced an objective response rate of 35% with a median PFS of 16.1 months, and a median OS of 27.2 months. Disease stability was seen in 38% of patients [50]. A follow-up pharmacokinetics/pharmacodynamics (PK/PD) analysis suggested that the efficacy of the drug in reducing tumor size is seen in patients with higher drug concentrations in plasma. The study in a larger cohort should be conducted given the efficacy and the tolerability of axitinib [51].

#### 4.1.2. BRAF-Targeting TKIs

*BRAF*V600E mutation is the most common somatic mutation in thyroid cancer of follicular cell origin. Thus, BRAF kinase inhibitors are expected to be effective against the *BRAF* mutated iodine-refractory thyroid cancers. Vemurafenib, a potent BRAF kinase inhibitor, has been approved by the FDA and the EMA for the treatment of late-stage melanoma [52]. An open-label, non-randomized, phase 2 trial at ten different centers worldwide, carried out between 2011 and 2013 in patients at the age of 18 or older with recurrent or metastatic, iodine-refractory, *BRAF*V600E-positive PTC, showed promising antitumor activity [53]. Partial response was observed in 38.5% of patients who were TKI naïve and 27.3% of patients who were previously treated with multi-kinase inhibitors. The median duration of response in the TKI-naïve cohort and in those who were previously treated with multi-kinase inhibitors was 16.5 and 7.4 months, respectively. Dabrafenib is a BRAF inhibitor studied in a large phase I clinical trial in patients with metastatic *BRAF*V600E-mutant thyroid cancer. Skin papillomas, alopecia and hyperkeratosis were the most common treatment-related adverse events [54]. The landmark study by Subbiah et al. demonstrated an unprecedented efficacy of dabrafenib (150 mg twice daily) and trametinib (MEK inhibitor, 2 mg trametinib once daily) combination therapy in patients with *BRAF*V600E-positive ATC [55]. The treatment efficacy of dabrafenib and trametinib in patients with *BRAF*V600E-mutated ATC in a phase II open-label clinical trial was remarkable, with a 69% overall response rate. The estimated 12-month overall survival was 80%, compared to the historical 12-month overall survival of 20–40%. Thus, FDA approved the combination of dabrafenib and trametinib for ATC with *BRAF*V600E mutation in 2018.

#### 4.1.3. MEK-Targeting TKIs

Mitogen-activated protein kinase (MAPK) kinase (MEK) comprises the several enzymes upstream to the MAPK signaling pathway. These enzymes selectively phosphorylate serine/threonine and tyrosine residues of the MAPK substrates. MEK plays a critical role in RAS/RAF-mediated carcinogenesis. Trametinib, an inhibitor of MEK1/2, has gained approval for clinical use over the past decade. Based on the remarkable durable efficacy seen in a phase II clinical trial (NCT02034110), the FDA approved the use of trametinib (MEKINIST^®^) in combination with dabrafenib (TAFINLAR^®^) in 16 patients with locally advanced or metastatic ATC with *BRAF* V600E mutation. The treatment with trametinib and dabrafenib resulted in an overall response rate of 69% (1 and 10 patients with complete and partial response, respectively) [55]. Selumetinib is another potent selective inhibitor of MEK1/2 that decreases RAS, RAF, and/or MAPK activity. Because radioactive iodine is an effective targeted therapy in advanced thyroid cancer of follicular cell origin, selumetinib was studied in 20 patients with advanced, iodine-refractory thyroid cancer with the goal to increase the uptake of radioiodine in thyroid cancer after treating the patients with selumetinib for 4 weeks. Iodine-124 positron-emission tomography (PET), before and after treatment with selumetinib (75 mg twice daily) showed enhanced uptake of iodine-124 in 12 out of 20 patients, with acceptable treatment-related toxicities. Of 12 patients with increased radioiodine uptake, 5 out of 5 patients had *NRAS-*mutated thyroid cancer, while 4 of 9 patients had thyroid cancer with *BRAF* V600E mutation. Eight patients with increased radioiodine uptake did receive therapeutic doses of radioiodine ablation with five confirmed partial responses [56]. In a follow-up phase II study completed in 2017, though the drug was well tolerated, it did not show an improvement in the primary outcome of radioiodine uptake in PTC patients [57].

#### 4.1.4. Anti EGFR TKIs

Gefitinib is an oral TKI with antineoplastic efficacy in metastatic cancers by inhibiting EGFR activation. In a phase II clinical trial of gefitinib in patients with radioiodine-refractory, locally advanced, or metastatic MTC and ATC, patients were treated with 250 mg of gefitinib per day. Although 32% of patients had an objective reduction in tumor size, none met the criteria for a partial response. The median PFS and OS were 3.7 and 17.5 months, respectively [58]. Due to lack of efficacy, gefitinib is not currently used in patients with advanced thyroid cancer.

Cetuximab is a recently developed chimeric monoclonal antibody targeting the ligand-binding domain of the EGFR. The FDA approved the use of cetuximab in *K-ras* wild-type, EGFR-expressing metastatic colorectal cancer and squamous cell cancer of head and neck. In a preclinical study using orthotopic ATC xenografts, cetuximab alone or in combination with irinotecan exhibited in vivo anti-proliferative effects [59]. However, there is no clinical trial of cetuximab in patients with thyroid cancer.

#### 4.1.5. mTOR Targeting TKIs

Derived from the immunosuppressive macrolide rapamycin, everolimus, and temsirolimus are small molecule inhibitors that cause an anti-proliferative effect by blocking the mechanistic target of rapamycin (mTOR) [60]. Being a member of the PI3K-related kinase family, the mTOR is a ubiquitously expressed protein kinase that phosphorylates its substrates at serine/threonine residue. A phase II clinical trial of everolimus (10 mg daily) was conducted in patients with progressive metastatic or locally advanced radioactive refractory DTC and ATC. About 65% of patients showed stable disease, and the median PFS and OS were 9 and 18 months, respectively. However, bone metastasis affected survival negatively [61]. The lack of efficacy was demonstrated in another phase II study involving patients with locally advanced or metastatic thyroid carcinoma. There was only one patient with the initial near-complete response after receiving 10 mg daily of everolimus for 18 months. This patient subsequently developed a progressive disease [62]. Later the whole genome sequencing revealed that the resistance to the drug was conferred by a nonsense mutation in TSC2. Most of the current clinical trials on everolimus include combinatorial studies with various drugs, such as sorafenib, pasireotide, lenvatinib, for ATC. Combination of temsirolimus and sorafenib in patients with recurrent or metastatic radioactive iodine-refractory thyroid cancer (*N* = 38) in a phase II trial showed improved partial response and PFS [63]. Because of a lack of promising efficacy, the mTOR inhibitors have not been clinically used in the treatment of advanced thyroid cancer.

### 4.2. Vascular Disrupting Agents

Thyroid cancer is a well-vascularized tumor, like most endocrine cancers. Because solid cancers largely depend upon tumor vasculature to supply nutrients and oxygen to survive and thrive, the use of vascular disrupting agents (VDAs) to reduce the tumor’s blood supply and alter the tumor microenvironment can lead to improved treatment efficacy. Combretastatin A4 phosphate (CA4P), is a VDA that targets the tumor microenvironment by depolymerizing the microtubules, disrupting the vascular endothelial cells, and reducing blood supply to the tumors. The safety and efficacy were demonstrated in the ATC cell lines and were further tested in phase I and II trials [64]. Crolibulin, a derivative of combretastatin, inhibits the polymerization of tubulin in a similar manner, with a better response in ATC. A phase I clinical trial of cisplatin and dose-escalating crolibulin was conducted in patients with advanced solid cancers, including recurrent or metastatic ATC. Of 16 patients with ATC, there was one patient with a complete response and one with stable disease. The most common grade 3 toxicities among the 21 enrolled patients were lymphopenia (33%), hyponatremia (29%), anemia (19%), hypertension during infusion (14%), and hypophosphatemia (9%) [65]. Though well tolerated, the combination would need further evaluation to be used as a regimen for ATC. Efatutazone (CS-7017), a synthetic peroxisome proliferator-activated receptor (PPAR) agonist, is a third-generation drug of the class thiazolidinedione and thus a potent stimulator of PPARγ-mediated transcriptional activation. The inhibitor demonstrated remarkable antitumor activity in the preclinical ATC model, while in a phase I clinical trial of efatutazone (0.3 mg) and paclitaxel in patients with ATC, safety and tolerance were shown in combination with paclitaxel [66]. The efficacy of this regimen was modest with disease stability observed in seven patients, while one patient showed partial response among the 15 enrolled patients. Patients who received 0.3 mg efatutazone had median survival of 138 days, and the median time to disease progression was 68 days. Common serious adverse events included anemia and localized edema, or fluid retention/edema.

### 4.3. Immunotherapy

An increasing number of studies have reported programmed death-ligand 1 (PDL1), as a key checkpoint regulator that alters the function of T cells after antigen-mediated stimulation. The PDL1 inhibitors are effective in reducing immune suppression in ovarian cancer, lung carcinoma, and melanoma, with increased survival in patients. Some of the immune checkpoint inhibitors include pembrolizumab, nivolumab, cemiplimab, avelumab, atezolizumab, and durvalumab [67]. In recent studies, PD-1 was found to be highly expressed in advanced DTC and ATC. Some of these inhibitors, such as pembrolizumab and nivolumab, are actively being studied in clinical trials (phase II) in combination with other small molecule inhibitors for ATC, PTDC, and MTC. Pembrolizumab is a PD-1-targeting humanized IgG4 antibody, initially approved by the FDA for metastatic melanoma and NSCLC, squamous cell carcinoma, head and neck cancer, and metastatic triple-negative breast cancer. A novel indication for the drug was approved in 2017, which included solid tumors with microsatellite instability (MSI-H) or mismatch repair (dMMR). Later in 2020, tumor mutational burden high (TMB-H) was further added as a new indication [67]. Similarly, CTLA-4, another immunosuppressive marker, was also considered as an important target owing to its negative correlation to thyroid differentiation and immunosuppressive markers. Two new drugs, durvalumab and tremelimumab are currently in early phase I trials in combination with RAI metastatic thyroid cancers and ATCs and were well tolerated. Regression of tumors was observed in 6 out of the 11 patients (50%) upon treatment with durvalumab plus RAI, two patients showed partial response, seven stable disease and two progression of disease as the best overall response. Six patients had tumor regression and four received treatment for more than 6 months [68]. Importantly, a recent clinical study revealed heterogeneity in the PDL-1 expression distribution among the subtypes of thyroid cancer. The expression of PDL-1 was also linked to clinicopathological features, such as mutations of *TERT* promoter and *BRAF* showing progression of advanced thyroid cancer [69].

## 5. Potential Advances in Future Therapies

### Nanomedicine Mediated Targeted Therapy

Nanomedicine has made an impact in clinical diagnostics and treatment with a promise to overcome limitations to drug delivery, by allowing agents that can selectively and preferentially deliver their payload to the tumor tissues. The high volume to surface ratio properties of nanoparticles ensures efficient targeted delivery of drugs to the targeted site to increase the efficacy of the drug. Nanoparticle-mediated targeted drug delivery in thyroid cancer was observed as a successful therapeutic strategy in some of the thyroid cancer clinical studies (Table 2). It is well known that the tumor necrosis factor alpha (TNF-α) receptors, folate receptor (FA), and thyroid-stimulating hormone receptor are highly expressed in thyroid cancer. In a recent study by our laboratory, a novel nanomedicine carrying recombinant human tumor necrosis factor-alpha (TNF-α) was developed that can be effectively used to reduce tumor IFP to increase paclitaxel delivery and anti-tumor efficacy [70]. Paclitaxel prodrug-loaded TNF-α conjugated gold nanomedicine improved the drug delivery in metastatic thyroid cancers and pancreatic neuroendocrine cancers. A recent study by another group demonstrated increased targeted delivery of doxorubicin, mediated by silicon dioxide nanoparticles (NPs) conjugated with thyroid stimulating-hormone receptor-targeting ligand resulting in a reduction in tumor size and off-target effects in a thyroid cancer in vivo model [71]. However, targeting thyroid stimulating hormone receptors is likely not effective in PDTC and ATC because these dedifferentiated tumors have low or no expression of thyroid stimulating hormone receptors. Mesoporous silica NPs conjugated with a folate antagonist, methotrexate (MTX), as a targeting ligand have been used for targeted delivery of fingolimod in thyroid cancer. The study showed that the MTX conjugated mesoporous silica NPs inhibit cellular proliferation and invasion in thyroid cancer both in vivo and in vitro and reduce the toxicity of the free drug cocktail [72]. Transferrin-modified hollow mesoporous silica nanoparticles (HMSNs) have been used for the targeted delivery of sorafenib in drug-resistant thyroid cancer TPC-1 and BCPAP cells. Sorafenib-encapsulated transferrin hollow mesoporous NPs induced more apoptosis through the RAF/MEK/ERK signaling pathway as compared to only sorafenib-encapsulated hollow mesoporous nanoparticles [73]. Among other nanoparticle-mediated drug delivery in thyroid cancer, polydopamine gold–silver alloy (Au–Ag^@^PDA NPs) plays a role in the mitochondrial regulation of the tumor cells. Au–Ag^@^PDA NPs can accumulate in the mitochondria, leading to mitochondrial dysfunction with downregulation of dihydroorotate dehydrogenase resulting in an upregulation of p53 and cell cycle arrest, ultimately inhibiting the proliferation of thyroid cancer cells [74]. More recently, the use of nanoparticles for thyroid cancer imaging has been also explored; for instance, Src homology 2-containing phosphotyrosine phosphatase 2 (SHP2)-targeted NPs have been used for a molecular probe for thyroid cancer [75]. An advanced study developed a novel theragnostic nanoplatform for controllable doxorubicin delivery in thyroid cancer [76]. The controllable doxorubicin drug delivery in thyroid cancer depends on pH, enzyme, and photothermal-mediated stimulation [76]. The study was the first to report nanoparticle-mediated photoacoustic imaging-guided chemo-photothermal therapy for thyroid cancer. In addition to the nanoparticle-mediated targeted drug delivery, the advanced RNAi nanotechnology has been recently developed for thyroid cancer therapy. A dual-purpose theragnostic NP platform composed of near-infrared (NIR) fluorescent polymers has been studied in preclinical thyroid cancer models: (i) siRNA delivery to effectively silence oncogene V-Raf murine sarcoma viral oncogene homolog B (BRAF) in tumor tissue, (ii) tumor accumulation tracking by tumor imaging in thyroid cancer in an in vivo model [77]. In the current scenario, the concept of nanoparticle-mediated drug delivery for improvement of efficiency and efficacy of the existing therapies seems to be a new paradigm shift in treating solid cancers, including thyroid cancer.

Intensive preclinical and clinical investigations have developed promising small molecule inhibitors that target cytosolic and membrane signaling molecules and kinases. Among the recently investigated TKIs, Cediranib has been demonstrated to block tumor growth and to prolong survival in an ATC murine xenograft model showing high specificity for VEGFR1 and VEGFR2 inhibition resulting in an anti-angiogenic effect. The compound mainly activated VEGFR2 and downstream signaling in endothelial cells, inhibited the proliferation of endothelial cells, and induced tumor cell apoptosis both in vitro and in vivo. Similarly, TERT promoter mutations are prevalent in PTC, with new hotspot mutations reported recently in the patients with or without *BRAF* V600E mutation, and therefore upcoming studies have described the mutations as potential targets for therapy [78].

Another promising compound is bortezomib, a protease inhibitor, that upregulates the proapoptotic protein NOXA (also known as phorbol-12-myristate-13-acetate-induced protein 1 (PMAIP1)), resulting in cell death. Bortezomib potentially suppresses the NF-κB signaling pathway, resulting in the downregulation of its anti-apoptotic target genes.

Apart from proteasomal and immune components, metabolic intermediates are highly involved in the progression of the tumors and can serve as promising targets for therapy. The Warburg effect [79], which marks the shift from the oxidative to the reductive type of metabolic phenotype, is known to be the hallmark of cancer cells. Metabolic pathway intermediates and enzymes have become the upcoming potentially promising candidates as distinct metabolic profiles have been observed in the subtypes, including ATC and metastatic thyroid cancer tissues. The upregulation of glycolytic intermediates and membrane transporters, such as glucose transferase (GLUT1), hexokinase 1 (HK1), hexokinase 2 (HK2), Phosphofructokinase 1 (PFK1), Phosphofructokinase 2 (PFK2), enolase (ENO), Lactate dehydrogenase A (LDHA), and Monocarboxylate transporters (MCTs) has been highlighted in a few studies in thyroid tumors [80,81]. Hypoxia-inducible factor 1 (HIF-1) is also activated by inflammatory processes, energy deprivation, and oxidative stress in undifferentiated thyroid cancers. Among the top anti-metabolic drugs that are approved or in clinical trials are methotrexate, pemetrexed, 5-fluorouracil, CB-839, AG-120, BAY1436032, AG221, AG-881, and AZD3965 [82]. Nevertheless, metabolic dependencies vary among the type of cancer and are highly influenced by the genetic alterations and microenvironment of the tumor; hence, the effectivity of these drugs was observed only in a subset of cancer population.

## 6. Conclusions

To date, enormous development has been made towards the molecular understanding of thyroid cancer and its subtypes, which has led to advancement in its therapeutic strategies. Though the prognosis of DTC is encouraging, PDTC and ATC show a poor prognosis because of the resistance to current therapies, and high rates of recurrence and metastases. The progress of preclinical and clinical investigations needs to continue until the discovery of a novel and effective therapeutic approach that significantly reduces thyroid cancer-related morbidity and mortality. Further investigations on the upcoming drug screening and therapies are warranted to improve the treatment modalities of thyroid cancer.

## Figures and Tables

**Table 1 cancers-13-06194-t001:** Recent clinical trials (2020–2021) conducted in thyroid cancer.

S No.	Phase	Drugs	Type of Thyroid Cancer	Target	Clinical Trial	Status
1	Phase 3	Selpercatinib + Cabozantinib + Vandetanib	MTC	MET, RET, VEGFR1/2/3, FLT3, KIT, TRKB, AXL	NCT04211337	Recruiting
2	Phase 3	Pralsetinib + Cabozantinib + Vandetanib	RET-mutated MTC	Mutated-RET, VEGFR1/2/3, FLT3, KIT, TRKB	NCT04760288	Not yet recruiting
3	Phase 2	Cabozantinib (40 mg)	ATC	MET, RET, VEGFR1/2/3, FLT3, KIT, TRKB	NCT04400474	Recruiting
4	Phase 2	Dabrafenib + Trametinib + PDR001	FTC, PTC	BRAF, MAP2K; MAPK/ERKK, MEK1/2	NCT04544111	Recruiting
5	Phase 2	Lenvatinib	DTC, ATC	VEGFR2	NCT04321954	Not yet recruiting
6	Phase 2	Lenvatinib + Pembrolizumab	mTC, PDTC, Stage IVB ATC,	VEGFR2, mutant and fusion products of RET	NCT04171622	Not yet recruiting
7	Phase 2	Cemiplimab + Dabrafenib + Trametinib	BRAF-V600 mutated ATC	BRAF, ME2K, MEK1/2	NCT04238624	Recruiting
8	Phase 2	Dabrafenib + Trametinib	RAI-refracted FTC	BRAF, ME2K, MEK1/2	NCT04554680	Recruiting
9	Phase 2	Dabrafenib + Trametinib	BRAF-ATC	BRAF, ME2K, MEK1/2	NCT04739566	Recruiting
10	Phase 1	Dabrafenib + Trametinib + IMRT *	BRAF-V600Kmutated ATC	BRAF, ME2K, MEK1/2	NCT03975231	Recruiting
11	Phase 1	Cabozantinib + Nivolumab	Advanced PTC	MET, RET, VEGFR1/2/3, FLT3, KIT, TRKB, PD-1, PD-L1, PD-L2	NCT04514484	Recruiting

* Intensity modulated radiation therapy.

**Table 2 cancers-13-06194-t002:** List of nanoparticles used for efficient delivery of targeted inhibitors.

Types of Nanoparticles (NPs)	Purpose	References
Gold NPs	Targeted delivery of TNF-alpha	[65]
Silicon dioxide NPs	Targeted delivery of Doxorubicin	[66]
Fingolimod and methotrexate loaded mesoporous silica NPs	Targeted delivery of Fingolimod	[67]
Transferrin encapsulated mesoporous silica NPs	Targeted delivery of sorafenib	[68]
Gold–silver alloy NPs	Mitochondrial dysfunction	[69]
Perfluoropentane encapsulated polylactic-co-glycolic acid-based nanoparticles (NPs)	Molecular probe of thyroid cancer	[70]
Gelatin-stabilized polypyrrole NPs	Controllable drug delivery of Doxorubicin	[71]
Near-infrared fluorescent nanoplatform	Imaging and systemic siRNA delivery to metastatic anaplastic thyroid cancer	[72]

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
