# Peer review of "Advances in Biomarker-Driven Targeted Therapies in Thyroid Cancer"

_cancers, 2021, doi:10.3390/cancers13246194_

Round 1
Reviewer 1 Report
This review by Dr Mishra and co-authors is a nice summary of thyroid cancer genetics and actionable mechanisms of clinical use. I found the manscript well-written with a clear clinical angle to it, and the literature is covered well for most parts. Tables are detailed, and makes up for the lack of figures. I have some suggestions for the authors to consider:
1. I assume this review focuses on thyroid cancer in adults, but I think it could be important to also mention pediatric thyroid cancer in a few passages of text - not least the overrepresentation of fusion gene events in pediatric PTC (which may be targetable if RAI fails), the notion of DICER1 miRNA regulator mutations in pediatric follicular thyroid tumors and PDTCs (again, with potential for future drugs targeting a faulty miRNA machinery).
2. The role of TERTp mutations in poor-prognosis PTC/FTC/FTUMP/PDTC/ATC may be elaborated in the "future" section, not least since there has been recent advances in the field of targeting these tumors (PMID: 33836600).
3. It is not clear to me if the authors advocate that all thyroid cancers should be interrogated by NGS in a clinical setting - what are the current recommendations from ATA/ETA? This review could help the reader to not only understand what drugs to use, but also when to ASK for molecular.
4. The immunotherapy section is vital to this review as it it THE hot topic of 2021. In this aspect, the authors could also take into account the recent discovery of mutations in immune-regulatory genes in high-risk pediatric PTCs, further demonstrating the need for subsets of thyroid cancers to interact and inhibit the immune system. Also, it seems as though the authors did not cite the relevant PD-L1 IHC papers?
5. Rows 74-75: "...mutations in other receptor tyrosine kinases (12%) such as RET, NTRK and ALK[7]." Do the authors mean fusions rather than mutations?
6. Rows 353-354: Please correct the row break.
Author Response
#Reviewer 1
This review by Dr Mishra and co-authors is a nice summary of thyroid cancer genetics and actionable mechanisms of clinical use. I found the manuscript well-written with a clear clinical angle to it, and the literature is covered well for most parts. Tables are detailed and makes up for the lack of figures. I have some suggestions for the authors to consider:
Response: We thank reviewer for the supportive comments and included information in our manuscript based on the suggestions made.
- I assume this review focuses on thyroid cancer in adults, but I think it could be important to also mention pediatric thyroid cancer in a few passages of text - not least the overrepresentation of fusion gene events in pediatric PTC (which may be targetable if RAI fails), the notion of DICER1 miRNA regulator mutations in pediatric follicular thyroid tumors and PDTCs (again, with potential for future drugs targeting a faulty miRNA machinery).
Response: Thank you for the suggestion. Additional information on the molecular genetics of pediatric thyroid cancers has been included in the sections 2.1, lines 96-101, and 2.2, lines 126-131.
- The role of TERTp mutations in poor-prognosis PTC/FTC/FTUMP/PDTC/ATC may be elaborated in the "future" section, not least since there has been recent advances in the field of targeting these tumors (PMID: 33836600).
Response: We agree that some recent advances are currently ongoing in the TERTp mutations, hence we added some information in the “future’ section 5.1, lines 560-563.
- It is not clear to me if the authors advocate that all thyroid cancers should be interrogated by NGS in a clinical setting - what are the current recommendations from ATA/ETA? This review could help the reader to not only understand what drugs to use, but also when to ASK for molecular.
Response: Thank you for the suggestion. The language regarding NGS and the current ATA guidelines has been added to the introductory paragraph of section 2, lines 66-69.
- The immunotherapy section is vital to this review as it it THE hot topic of 2021. In this aspect, the authors could also take into account the recent discovery of mutations in immune-regulatory genes in high-risk pediatric PTCs, further demonstrating the need for subsets of thyroid cancers to interact and inhibit the immune system. Also, it seems as though the authors did not cite the relevant PD-L1 IHC papers?
Response: Thank you, we have added some more literature in section 4.3 as suggested and included a relevant PD-L1 IHC paper in the manuscript. Please see lines 496-499.
- Rows 74-75: "...mutations in other receptor tyrosine kinases (12%) such as RET, NTRK and ALK [7]." Do the authors mean fusions rather than mutations?
Response: Thank you for the feedback. We have changed the language in this section, please see line 84-85.
- Rows 353-354: Please correct the row break.
Response: This has been rectified now.
Reviewer 2 Report
The review presented by Mishra and colleagues about the “Advances in Biomarker-Driven Targeted Therapies in Thyroid Cancer” is an interesting collection of novel drug treatment and delivery methods in thyroid cancer.
The work suitable describes the issue, and it results well organized.
Minor revision:
- lines 45 – 49: the authors have to affirm also in this paragraph that not all WDTC respond in the same way to the RAI therapy. The diagnosis of a WDTC does not include the response to RAI treatment.
- line 114: the two MEN syndromes need to be presented before addressing the characteristic of each of them.
- The work needs a spell check (e.g., line 246, the word “cacner”) and a style check (e.g., line 399 – 407, grey color)
Author Response
#Reviewer 2
Comments and Suggestions for Authors
The review presented by Mishra and colleagues about the “Advances in Biomarker-Driven Targeted Therapies in Thyroid Cancer” is an interesting collection of novel drug treatment and delivery methods in thyroid cancer.
The work suitable describes the issue, and it results well organized.
Response: We appreciate for the reviewer’s positive feedback and minor comments, the responses to the comments are as mentioned below.
Minor revision:
- lines 45 – 49: the authors have to affirm also in this paragraph that not all WDTC respond in the same way to the RAI therapy. The diagnosis of a WDTC does not include the response to RAI treatment.
Response: Thank you for the comment, the language has been modified to reflect this point. Please see page 2, line 49.
- line 114: the two MEN syndromes need to be presented before addressing the characteristic of each of them.
Response: Thank you for the feedback. A thorough description of MEN2A and MEN2B is currently present in the text of this section.
- The work needs a spell check (e.g., line 246, the word “cacner”) and a style check (e.g., line 399 – 407, grey color)
Response: The error has been rectified.